# Theoretical and Experimental Study of Friction Characteristics of Textured Journal Bearing

**DOI:** 10.3390/mi14030577

**Published:** 2023-02-28

**Authors:** Hongtao Wang, Wenbo Bie, Shaolin Zhang, Tengfei Liu

**Affiliations:** 1College of Mechanical and Power Engineering, Zhengzhou University, Zhengzhou 450001, China; 2Henan Province Engineering Research Center of Ultrasonic Technology Application, Pingdingshan University, Pingdingshan 467000, China

**Keywords:** textured journal bearing, tribological properties, laser etching, ultrasonic vibration processing

## Abstract

The proposed lubrication theory of textured journal bearing is a major innovation in the study of the tribological properties of surface morphology. When it comes to the study of surface topography, it is essential to consider the effect of surface roughness when analyzing the characteristics of journal bearing. In this paper, a Reynolds equation containing longitudinal roughness is established for journal bearing and solved by the finite difference principle to obtain the bearing load and friction characteristics. Subsequently, a combination of laser etching and ultrasonic vibration milling processes was used to prepare 5 µm, 20 µm, and 40 µm bearing friction subsets with square micro-texture surfaces. The analysis of the results shows that the surface roughness distributed in the non-texture region can substantially increase the oil film pressure. When the roughness profile and the surface weave work together, the presence of a surface texture with an optimum depth of 5 µm within a roughness range of less than 1.6 µm can improve the load-bearing characteristics by a maximum of 43%. In the study of the preparation of textured bearing friction substrate, it was found that laser etching can ablate the surface of the friction substrate to a depth greater than 20 µm with the ideal effect, while the surface texturing to a depth of 5 µm is more suitable using an ultrasonic vibration processing process. In the simplified journal bearing operating condition, the frictional wear test shows that if the effect of roughness is considered, the frictional force of the depth of 20 µm and 40 µm is significantly reduced and changes less with increasing load, while the frictional force of the textured frictional pair with a depth of 5 µm is improved but significantly affected by the load carrying capacity. Therefore, when the difference between the roughness profile and the depth of the texture is of a small order of magnitude, it indicates that the effect caused by the roughness factor is not negligible.

## 1. Introduction

The rapid development of science brought the knowledge of tribology to a new level. With the continuous improvement of fluid dynamic pressure and tribological theory, achieved by scholars in anti-wear, friction reduction research is also more in-depth [1,2]. As a result, the surface morphology of bearing friction subsets gradually entered the field of exploration by searchers. The improvement of tribological performance by changing the surface morphology of frictional subsets [3] is widely used in the fields of medicine, aviation, and mechanical equipment [4]. In-depth investigation of the influence of surface morphology on the performance of journal bearing is not only the need to explore the theory of journal bearing in fluid lubrication, but also the urgent needs of production practice.

Journal bearing friction pairs usually require a lubricating medium to reduce friction. In traditional tribological studies, the smoother the frictional substrate, the more favorable it is often considered in the fluid lubrication state [5,6]. In recent years, it was found that having a certain specific surface profile can instead improve tribological properties [7]. Usually, a specific uneven surface is academically known as surface texture of the friction surface [8], and surface texture is an effective means to improve the tribological properties, which can make the material surface achieve a self-lubricating effect and can reduce the premature failure and energy consumption of mechanical equipment caused by frictional wear [9], and can effectively improve the bearing load capacity and reduce the friction coefficient [10]. Then, during the texturing of the friction substrate, the geometric parameters of the texture and the location of the distribution [11] affect the bearing characteristics.

In the case of dynamic pressure bearings, for example, the weave is often distributed in the lift zone of the lubricant film [12]. In order to prepare the ideal textured bearing, several options are available, among which, laser surface technology, X-ray lithography [13], reactive ion etching [14], embossing [15], and CNC vibratory processing [16] are the most common processing methods.

Most scholars analyzed and experimentally verified the theoretical models [17,18,19] for weave-configured bearings and concluded that the existence of optimal texture parameters [20] can improve the bearing performance, while in practice, the non-texture regions in the microscopic state are not smooth, and there are successively different micro-convex fronts and pits on the surface, and these are the surface roughness. In order to be closer to the real working conditions, the surface roughness of the prepared textured bearing cannot be neglected in the theoretical calculations. With the surface roughness as a global surface topography, the local effect from the contact gap that varies with the roughness can increase the friction coefficient, and the global roughness starting at the boundary position can have a gain effect on the sliding bearing [21,22]. Moreover, the surface texture as a local surface morphology, together with the surface roughness profile, affects the friction characteristics of the bearing.

Thus, this paper establishes the theoretical model of the square textured bearing and couples the roughness-influencing factors, calculates the static and dynamic characteristics of the bearing, and performs the experimental analysis of the friction characteristics. The Reynolds equation is solved by a finite difference method, and the abrupt change in film thickness in the texture region is bounded to analyze the effect of lubrication characteristics for radial dynamic pressure plain bearings that are smooth, contain a single roughness, a single micro-texturing, coupled roughness, and two surface morphologies of texturing. On a theoretical basis, it was found that surface finishing, such as milling, boring, grinding, and lapping of the bearing surface, is highly correlated with the bearing surface morphology, as influenced by the machining accuracy [23]. In order to obtain uniform and neat bionic lattices, textures and other structures on the surface of friction subsets, the processes of ultrasonic vibration milling [24] and laser etching of textured bearing friction subsets have a promising future in bearing tribology research [25].

## 2. Theoretical Model and Experimental Method

### 2.1. Theoretical Model

Texture bearing, that is, in a certain area of the friction sub-surface to prepare a regular arrangement, fixed the size of micro-pits. In this paper, all regularly arranged square micro-textures are used. Figure 1a shows the schematic diagram of the axial direction of the textured dynamic pressure bearing. Figure 1b shows the radial structure of the bearing. In the picture, *φ* is the circumferential coordinate, *λ* is the axial coordinate, *y* is the radial (film thickness direction) coordinate, *e* is the eccentricity distance, *Ω* is the angular velocity, and *θ* is the eccentricity angle. The square texture is evenly distributed on the surface of the friction substra@te, and *ξ* indicates the zone of texture.

Figure 2a,b represent the schematic diagram of the lubricant film in the smooth state. In the figure, *h* indicates the lubricant film thickness in the smooth state, and *h_t_* indicates the amount of change in film thickness caused by the weave depth, and the ratio of *h_t_* to radius gap *c* is defined as the dimensionless texture depth *h_t_*. If the lubricant film thickness is expanded along the maximum film thickness, its two-dimensional view is shown in Figure 2c, where the square textures are all sequentially arranged in the lubricant film pressure ramping area.

Figure 3 shows the microscopic diagram of the film existing in the bearing gap after adding longitudinal roughness. In this model, the roughness is assumed to have the form of long, narrow ridges and valleys running in the direction of sliding, and the x-direction is considered smooth. As shown in Figure 3a, the longitudinal direction contains the roughness that conforms to the Gaussian distribution. *Ra* and *h_t_* essentially belong to the surface morphology, *h_t_* belongs to the local roughness, and *Ra* is a global surface morphology; both will have an impact on the oil film thickness, then the *Ra* rough peak caused by the lubricant film thickness change recorded as *h_s_*, such as Figure 3b, coupled roughness of the texture structured bearing actual lubricant film thickness for H.

### 2.2. Mathematical Model

Rough crests and troughs, as well as texture depth, are the main factors that cause changes in film thickness. Assume that the lubricant is a Newtonian fluid, ignoring thermal effects and that the lubricant temperature is constant. Roughness caused by the amount of film thickness change *h_s_* and texture depth caused by the amount of film thickness change *h_t_* are the main factors of the film thickness change, so take into account the roughness of the texture structured dynamic pressure bearing film thickness expression (*ε* is the eccentricity and *ξ* is the region where the texture exists):(1)$$\left\{ \begin{array}{ll} \bar{H}=1+\varepsilon \cos\varphi +{\bar{h}}_{s} & \varphi \notin \xi \\ \bar{H}=1+\varepsilon \cos\varphi +{\bar{h}}_{s}\;+{\bar{h}}_{t} & \varphi \in \xi \end{array}\right.$$

Figure 4 shows the assessment method of the surface roughness profile. Since surface roughness is the average profile value formed by rough peaks and rough troughs, the film is influenced by the rough profile. In order to establish a certain connection between roughness values and film thickness, the rise and fall of surface roughness peaks and valleys are treated as a random variable, i.e., when the random variable is constant, it means that the rough surface has a very similar roughness height distribution.

In order to make the simulated roughness height closer to the real value, find the statistical characteristic parameters (mathematical expectation) with the same surface roughness [26]. The bearing surface roughness is affected by the preparation process in the same direction, and it is assumed that only longitudinal roughness exists in this calculation example. Since the bottom surface of the machined texture is parallel to the bearing surface and the entire bearing surface roughness is in accordance with Gaussian distribution, the mathematical expectation expression of the probability density distribution of film thickness under the influence of roughness is:(2)$$E\left(h_{s}\right)=\int_{-\infty }^{\;\infty }h_{s}\;f\left(h_{s}\right)dh_{s}$$

In order to describe the output value of the random roughness, the probability function near some determined value point of the Gaussian roughness distribution is set as a Gaussian probability density function. The formula is:(3)$$f\left(x\right)=\frac{1}{\sqrt{2\pi }\sigma }\times e^{-\frac{{\left(x-\mu \right)}^{2}}{2\sigma^{2}}}$$

Usually, the roughness is assessed by selecting a length of the machined surface and using a profilometer to obtain the surface roughness profile. In most of the roughness profile distribution patterns, the roughness distribution and Gaussian distribution tend to be consistent. For analysis purposes, the Gaussian distribution is used instead of the amplitude density of the actual surface roughness. The calculation uses a polynomial distribution to simulate the Gaussian distribution of the random roughness, introducing the roughness parameter *C*. The equation is:(4)$$f\left(h_{s}\right)=\left\{ \begin{array}{ll} \frac{35}{32C_{}^{7}}{\left(C_{}^{2}-h_{s}^{2}\right)}^{3} & ,\;\left|h_{s}\right|<C\\ 0 & \;,\;\left|h_{s}\right|\geq C \end{array}\right.$$

In the above equation, *f*(*h_s_*) is the probability density distribution function of the independent variable *h_s_* for the change in film thickness caused by roughness; *C* denotes the roughness parameter, and *C* = 3*σ.*

There is a transformation relationship between the standard deviation of the Gaussian distribution and the roughness distribution of the polynomial distribution. The equation is:(5)$$\sigma^{2}=\int_{-\infty }^{\infty }x^{2}f\left(x\right)dx=\int_{-C}^{C}x^{2}\frac{35}{32C^{7}}{\left(C^{2}-x^{2}\right)}^{3}dx=C^{2}/9$$

In the actual machining process of the specimen, the surface quality of the machining is usually characterized using the arithmetic mean deviation *Ra* of the profile, so it is necessary to establish the conversion relation of *Ra* in the Reynolds equation. *Ra* is related to the roughness parameter *C* as:(6)Ra=E(x−E(x))=E(|x|)
(7)Ra=0.789021C/3

The dimensionless Reynolds equation is (*p* is the dimensionless film pressure):(8)$$\frac{\partial }{\partial \varphi }\left[E\left({\bar{H}}^{3}\right)\frac{\partial \bar{p}}{\partial \varphi }\right]+{\left(\frac{d}{l}\right)}^{2}\frac{\partial }{\partial \lambda }\left[\frac{1}{E\left(1/{\bar{H}}^{3}\right)}\frac{\partial \bar{p}}{\partial \lambda }\right]=3\frac{\partial \bar{H}}{\partial \varphi }$$

In this paper, the boundary conditions used in the numerical calculation process are:(9)$$\left\{ \begin{array}{ll} \lambda =\pm 1,\bar{p}=0 & \;\mathrm{Both}\;\mathrm{ends}\;\mathrm{of}\;\mathrm{the}\;\mathrm{bearing}\;\;\\ \varphi =0,\bar{p}=0 & \;\mathrm{Lubricant}\;\mathrm{film}\;\mathrm{starting}\;\mathrm{edge}\;\;\\ \partial \bar{p}/\partial \varphi =0,\bar{p}=0 & \;\mathrm{Lubricant}\;\mathrm{film}\;\mathrm{rupture}\;\mathrm{edge}\;\; \end{array}\right.$$

### 2.3. Tribological Characteristics

For the texturized bearing coupled with the roughness factor, the dimensionless fractional forces in horizontal and vertical directions are *Fx* and *Fy*, respectively, and the dimensionless bearing force is *F*. The formula is:(10)$$\left\{ \begin{array}{l} {\bar{F}}_{x}=-\int_{\Phi_{1}}^{\Phi_{2}}\left(\int_{-1}^{1}\bar{p}d\lambda \right)\sin\Phi d\Phi \\ {\bar{F}}_{y}=-\int_{\Phi_{1}}^{\Phi_{2}}\left(\int_{-1}^{1}\bar{p}d\lambda \right)\cos\Phi d\Phi \\ \bar{F}=\sqrt{{\bar{F}}_{x}{}^{2}+{\bar{F}}_{y}{}^{2}} \end{array}\right.$$

Friction in the bearing mainly consists of two parts, part of the lubricating medium and shaft and the shingle contact friction between them; the other part by the pressure flow resistance and shear resistance between the lubricant superimposed. The shear resistance between the lubricant film is calculated and superimposed by the shear resistance of the film integrity zone and the film rupture zone, respectively. The dimensionless expressions for the frictional pressure flow resistance and shear flow resistance of the texturization dynamic plain bearing is:(11)$$\left\{ \begin{array}{l} {\bar{F}}_{t1}=\frac{F_{t1}\psi^{2}}{\mu_{0}\Omega rl}=\frac{1}{2}\int_{-1}^{1}\int_{\Phi_{1}}^{\Phi_{p}}\frac{\bar{\mu }}{\bar{h}}d\Phi d\lambda +\frac{1}{2}\int_{-1}^{1}{\bar{h}}_{p}\int_{\Phi_{1}}^{\Phi_{p}}\frac{\bar{\mu }}{{\bar{h}}^{2}}d\Phi d\lambda \\ {\bar{F}}_{t2}=\frac{F_{t2}\psi^{2}}{\mu_{0}\Omega rl}=\frac{1}{2r}\left({\bar{F}}_{y}\varepsilon_{x}-{\bar{F}}_{x}\varepsilon_{y}\right) \end{array}\right.$$

The expressions for the frictional resistance to pressure flow and the frictional resistance to shear flow taking into account the surface roughness are:(12)$$\left\{ \begin{array}{l} {\bar{F}}_{t1}=\frac{F_{t1}\psi^{2}}{\mu_{0}\Omega rl}=\frac{1}{2}\int_{-1}^{1}\int_{\Phi_{1}}^{\Phi_{p}}\bar{\mu }\cdot E\left(\frac{1}{\bar{H}}\right)d\Phi d\lambda +\frac{1}{2}\int_{-1}^{1}{\bar{h}}_{p}\int_{\Phi_{p}}^{\Phi_{2}}\bar{\mu }\cdot E\left(\frac{1}{{\bar{H}}^{2}}\right)d\Phi d\lambda \\ {\bar{F}}_{t2}=\frac{F_{t2}\psi^{2}}{\mu_{0}\Omega rl}=\frac{1}{2}\bar{F}\varepsilon \sin\theta \end{array}\right.$$

### 2.4. Film Discretization

Figure 5 shows the mesh division of the bearing film. The film is expanded in the circumferential direction starting from the maximum thickness and divided into *n* grids along the axial direction, and the divided grids are numbered with *j*(*j* = 0~*n*), since *λ* the axial direction is indicated. The circumferential grid is uniformly divided into *m* cells, and each grid number is denoted by *i*(*i* = 0~m), since *φ* the circumferential direction is indicated. The step length per cell is: Δ*λ* = 2/*n*; Δ*φ* = (*φ* – *φ*)/*n.*

Figure 6 shows the application of the finite difference method on a single node. In order to calculate the pressure approximation at each point, a point (*i*,*j*) in the grid is taken for analysis, the pressure at point (*i*,*j*) is *p_i,j_*, and the first-order partial derivative can be approximated by the intermediate difference quotient, and the second-order partial derivative can be expressed jointly by the first-order partial derivative at a point with half of the adjacent steps and the value of the node with the same adjacent steps.

The expression for the second-order partial derivative at node (*i,j*) is:(13)$${\left[\frac{\partial }{\partial \varphi }\left({\bar{H}}^{3}\frac{\partial \bar{p}}{\partial \varphi }\right)\right]}_{i,j}\approx \frac{{\bar{H}}_{i+\frac{1}{2},j}^{3}{\bar{p}}_{i+1,j}+{\bar{H}}_{i-\frac{1}{2},j}^{3}{\bar{p}}_{i-1,j}-\left({\bar{H}}_{i+\frac{1}{2},j}^{3}{\bar{p}}_{i+1,j}+{\bar{H}}_{i-\frac{1}{2},j}^{3}{\bar{p}}_{i+1,j}\right){\bar{p}}_{i,j}}{{\left(\Delta \varphi \right)}^{2}}$$

The second order derivative of the film pressure in the circumferential and axial directions is found and the expression is:(14)$$A_{i,j}{\bar{p}}_{i+1,j}+B_{i,j}{\bar{p}}_{i-1,j}+C_{i,j}{\bar{p}}_{i,j+1}+D_{i,j}{\bar{p}}_{i,j-1}-E_{i,j}{\bar{p}}_{i,j}=F_{i,j}$$

The coefficients in the above equation are:(15)$$\left\{ \begin{array}{l} A_{i,j}={\bar{h}}^{3}{}_{i+\frac{1}{2},j}\\ B_{i,j}={\bar{h}}^{3}{}_{i-\frac{1}{2},j}\\ C_{i,j}={\left(\frac{d}{l}\cdot \frac{\Delta \varphi }{\Delta \lambda }\right)}^{2}{\bar{h}}^{3}{}_{i,j+\frac{1}{2}}\\ D_{i,j}={\left(\frac{d}{l}\cdot \frac{\Delta \varphi }{\Delta \lambda }\right)}^{2}{\bar{h}}^{3}{}_{i,j-\frac{1}{2}}\\ E_{i,j}=A_{i,j}+B_{i,j}+C_{i,j}+D_{i,j}\\ F_{i,j}=3\Delta \varphi ({\bar{h}}_{i+\frac{1}{2},j}-{\bar{h}}_{i-\frac{1}{2},j}) \end{array}\right.$$

### 2.5. Texture Boundary Film Thickness Variation Processing

Figure 7 shows the flow balance principle in the textured area. For textured bearing, considering the existence of step change in film at the texture, the film thickness abrupt change in the differential method is to be considered separately. Because the density of the divided grid is large and the weaving region does not completely overlap, the grid intersection must be on the texture, so the relationship between the flow balance at the abrupt change in film can be handled, and the *λ* and *φ* directions have a step change in film thickness. The density of micro-pits at a certain defined area is characterized by the micro-pit area rate *s*, which refers to the ratio of the circumferential cross-sectional area of a single micro-pit to the area of the control cell. The control cell is a virtual square (dashed line in the figure) enclosed by the midpoints of the spacing of adjacent micro-pits, e.g., the area ratio of a square micro-pit is *s* = (*a/m*)^2^, where the control cell side length is *m* and the side length of a square micro-pit is *a*.

When considering the flow balance of a single texture region, it is necessary to distinguish the film thicknesses on the upper and lower sides of row *j*. With *j* + 0 denoting the film thickness on the lower side of the *j* row infinitely close to the *j* row, and with *j* − 0 denoting the film thickness on the upper side infinitely close to the *j* row. In the formula, *d* is the bearing diameter, and *l* is the bearing width. *H* indicates the value of the film thickness at that point. The coefficients *A*, *B*, *C*, *D*, *E*, and *F* denote the coefficients of the pressure iterations. The coefficients of each of the textured bearings are:(16)$$\left\{ \begin{array}{l} A_{i,j}=\frac{{\bar{H}}_{i+\frac{1}{2},j+0}^{3}+{\bar{H}}_{i+\frac{1}{2},j-0}^{3}}{2}\\ B_{i,j}=\frac{{\bar{H}}_{i-\frac{1}{2},j+0}^{3}+{\bar{H}}_{i-\frac{1}{2},j-0}^{3}}{2}\\ C_{i,j}={\left(\frac{d}{l}\right)}^{2}{\bar{H}}_{i,j+\frac{1}{2}}^{3}\\ D_{i,j}={\left(\frac{d}{l}\right)}^{2}{\bar{H}}_{i,j-\frac{1}{2}}^{3}\\ E_{i,j}=A_{i,j}+B_{i,j}+C_{i,j}+D_{i,j}\\ F_{i,j}=\frac{3}{2}\Delta \varphi \left(\begin{array}{l} {\bar{H}}_{i+\frac{1}{2},j+0}+{\bar{H}}_{i+\frac{1}{2},j-0}\\ -{\bar{H}}_{i-\frac{1}{2},j+0}-{\bar{H}}_{i-\frac{1}{2},j-0} \end{array}\right) \end{array}\right.$$

Taking a point M on the grid as an example, the flow rate at that point according to the expression for flow balance is:(17)$$Q_{a}+Q_{b}+Q_{c}+Q_{d}+Q_{e}+Q_{f}=0$$

In the region Δ*φ* × Δ*λ*, a single weave is located in the divided film grid. In this equation, *Q* represents the flow rate of a control unit at a point on the grid, *U* represents the bearing speed, *μ* represents the lubricant viscosity, and *p* represents the film pressure. the expression for the flow rate through a single texture on this region is:(18)$$\left\{ \begin{array}{l} Q_{a}\approx \frac{\Delta z}{2}\left[\frac{Uh_{a}}{2}-\frac{h_{a}^{3}}{12\mu }{\left(\frac{\partial p}{\partial x}\right)}_{a}\right]\\ Q_{b}\approx \frac{\Delta z}{2}\left[\frac{Uh_{b}}{2}-\frac{h_{b}^{3}}{12\mu }{\left(\frac{\partial p}{\partial x}\right)}_{b}\right]\\ Q_{c}\approx \Delta x\left[-\frac{h_{c}^{3}}{12\mu }{\left(\frac{\partial p}{\partial z}\right)}_{c}\right]\\ Q_{d}\approx \frac{\Delta z}{2}\left[\frac{Uh_{d}}{2}-\frac{h_{d}^{3}}{12\mu }{\left(\frac{\partial p}{\partial x}\right)}_{d}\right]\\ Q_{e}\approx \frac{\Delta z}{2}\left[\frac{Uh_{e}}{2}-\frac{h_{e}^{3}}{12\mu }{\left(\frac{\partial p}{\partial z}\right)}_{e}\right]\\ Q_{f}\approx \Delta x\left[-\frac{h_{f}^{3}}{12\mu }{\left(\frac{\partial p}{\partial z}\right)}_{f}\right] \end{array}\right.$$

## 3. Methods and Tests

### 3.1. Solving Method

Figure 8 shows the flow chart for the numerical solution of the bearing characteristics. The Reynolds equation is solved by programming in C language. It is necessary to determine whether any point in the film belongs to the texture region and then proceed to the next step of calculation. When the convergence condition is satisfied, the bearing characteristic parameters are output.

### 3.2. Bearing Preparation

Figure 9 shows the structural parameters of the bearing friction pair. In order to facilitate the frictional wear test, the frictional subassembly of the sliding bearing is simplified, and the lubricant fills the minimum oil film thickness between the friction pairs. The bearing material is 42CrMo; 42CrMo is an ultra-high strength steel with high strength and toughness, and the sliding bearings made of this material are widely used in high-speed and heavy-duty rotating instruments. The test piece tested had a bearing radius of 25 mm, a shingle width of 40 mm, and a bearing width of 42 mm. the bearing surfaces are ground and finished to obtain test pieces with roughness contours on the surface, the initial gap size is adjusted to simulate a given eccentricity, and the eccentricity is changed by manual loading.

By means of laser marking, the required texture shape can be machined on the surface of the bearing specimen. Different texture depths can be obtained by changing the power and laser linear density of the marking machine. The processing principle is shown in Figure 10a, and the physical photo of the textured friction pair is shown in Figure 10b.

Figure 11 shows the three-dimensional shape of the surface texture formed by laser etching, the power parameter is set to 30% of the full-load power, and the laser beam linear density of 0.03 can obtain the surface texture with a depth of 20 μm. In the figure, due to the laser direction along the axial movement, in the axial direction, the texture wall is relatively smooth, the texture wall in the radial direction is not regular, and the bottom is not flat, but the overall square profile is shown. By setting the power parameter to 60% of the full power, a surface texture of 40 μm in depth can be obtained.

Figure 12 shows the three-dimensional shape of laser etching at a depth of 5 μm. Due to the limitations of the laser marking machine, laser etching does not work well at shallow texture depths. Figure 12a shows the surface texture at low power, where the molten metal rapidly condenses and accumulates inside the texture, and the actual depth does not reach 5 μm. Figure 12b shows the surface texture at low laser filling density, where the bottom contour of the texture forms a striated profile along the laser path. The surface fabric in Figure 12b shows a less dense laser filling, where the bottom profile of the fabric forms a striated profile along the laser path. Figure 12c shows a more dense laser filling, where the inner wall of the fabric is depressed and the bottom of the fabric forms a “volcanic” protrusion.

In order to achieve the machining of the surface texture with a depth of 5 μm, the surface of the friction subsets of the shaft shank was milled using an ultrasonic vibration process, and Figure 13 illustrates the ultrasonic vibration milling mechanism. Based on the Henfux-HFM700L CNC machining center, the surface of the bearing friction sub is woven. Under reasonable milling parameters, the high-frequency impact generated by longitudinal ultrasonic vibration forms regular micro-textures on the friction subsurface, which include surface roughness contours and surface texture.

Figure 14 shows the surface morphology formation mechanism of the test piece under ultrasonic milling. The bearing surface morphology was photographed using a COXEM EM-30plus benchtop SEM and a Bruker Nano Inc NPFLEX white light interference instrument. Figure 14a shows a physical view of the bearing tile, and Figure 14b shows the microscopic morphology there, where the superimposed trajectory of the knife blade movement forms a square surface texture. Figure 14c shows the roughness profile of the surface. The surface roughness profile of this friction sub is uniformly distributed and the depth is up to 5 μm, which is a more ideal preparation means.

### 3.3. Frictional Wear Test

The tribological characteristics of sliding bearings were analyzed by using a M2000 tribological wear testing machine, Figure 15 shows the physical diagram; the testing machine consists of three parts, respectively, the lubricant supply device, tribological wear device, and the data detection device. The maximum speed of the test machine is 400 r/min and the maximum applied load is 2000 N. During the test, the motor drives the lubricant in the reservoir along the oil path to the surface of the test piece, through the spindle rotation, the lubricating oil attached to the journal, and the shaft tile between the formation of dynamic pressure oil film. The controlled lubricating oil temperature is stable at 20 ℃, and finally with 200 r/min and 400 r/min for the test.

Figure 16 shows the schematic diagram of the processing principle of the friction characteristic test. If a woven bearing with roughness is not considered to be in a mixed friction state, i.e., assuming full fluid lubrication at all times, a steady flow of lubricant is provided in the reservoir. The spindle rotation brings the lubricant adhering to the bearing into the wedge gap and thus creates a dynamic pressure effect.

## 4. Analysis of Theoretical Results

### 4.1. Textured Bearing Characteristics Considering Roughness

The bearing parameters used in this calculation are shown in Table 1. The bearing characteristics were calculated and analyzed with a texture percentage of 36%. Figure 17 shows the film development of the textured bearing film, and Figure 17a,b shows the film thickness distribution for the textured depth of 5 μm and 20 μm, respectively. As the flow inside the surface texture is always kept in equilibrium, the film caused by the change in texture depth is peaked and increases with the depth of the texture, and the bearing capacity of the oil film adhering to the inner wall of the texture is reduced, so that the film becomes slender.

Figure 18 shows the comparison of film pressure considering each factor of surface topography. Figure 18a–d show the pressure distribution of film under the same working condition for bearings with a smooth surface, bearings with only roughness on the surface, bearings with only texture on the surface, and bearings with both roughness and texture on the surface, respectively. The peak dimensionless pressure in the figure is 0.464 for the smooth state and 0.470 for the textured bearing. The presence of the textured structure increases the film thickness, and thus the oil film pressure, so the most pressure peak is increased by 50% after coupling the two surface morphologies.

Figure 19 shows the trend of dimensionless bearing capacity with increasing eccentricity. When roughness is not considered, the surface texture with an eccentricity of 0.1~0.5 and depth of 10 μm has little effect on bearing load capacity. When the eccentricity is greater than 0.6, the bearing load capacity of the fabricated bearing decreases slightly. When the eccentricity is greater than 0.2, the bearing capacity is improved by considering the effect of roughness. Since the rough peaks on the shaft and shingle surfaces have a certain ability to trap lubricant, consideration of roughness will result in an increase in the load carrying capacity of the textured bearing.

Figure 20 shows the trend of dimensionless friction with increasing eccentricity. With small eccentricity under the account of surface roughness, bearing friction increases, and the larger the eccentricity, the more obvious the increase; the introduction of the texture of the texture can effectively reduce bearing friction. When the eccentricity is 0.9, the roughness and surface textures are introduced and the bearing friction increases significantly, indicating that the presence of textures and surface roughness under large eccentricity is not conducive to the bearing reducing friction.

### 4.2. The Relationship between Texture Depth and Roughness

As shown in Figure 21, at 0.4 eccentricity, the preparation of a certain regular texture on the surface of the frictional subsets of the dynamic bearing can lead to an increase in the dimensionless load carrying capacity, and the depth of the texture is 5 μm, which has the best effect on the load capacity improvement. Compared to bearings without texture, the load capacity can be increased by 2.3% at a texture depth of 15 μm and *Ra* = 1.3 μm. When the depth of the texture is greater than 15 μm, the presence of the texture has a negative effect on the dimensionless bearing capacity. If the joint effect of roughness and texture is considered and the roughness *Ra* = 0.26 μm (*C* = 0.01) is taken into account, the valley formed between its roughness peaks has a weak interception effect on the lubricant, and the dimensionless bearing capacity can be improved. The greater the value of roughness, the greater the increase in dimensionless bearing capacity.

Figure 22 shows the percentage increase in dimensionless load capacity due to the combined effect of roughness and surface texture for an eccentricity of 0.4. Compared with the dynamic pressure bearing without considering the surface roughness, if coupling the two surface morphologies of texture and roughness, the dimensionless load capacity improvement percentage is shown in the figure. Considering the effect of surface roughness, the dimensionless load capacity can be increased by up to 43% as the roughness increases. The improvement of dimensionless bearing capacity gradually decreases in the range of textured depth from *h_t_* = 5 μm to *h_t_* = 40 μm, but the smallest (*h_t_* = 40 μm, *Ra* = 0.26 μm) also has an improvement of 40.5% in order to optimize the surface roughness and surface texture for load-bearing capacity enhancement, as shown in the dark red and red and yellow areas on the way. During the processing of the textured bearings, the optimum depth of the prepared textures is *h_t_* = 5 μm to *h_t_* = 20 μm within a reasonable roughness range, as it is not possible to precisely control the surface roughness values.

Figure 23 shows the trend of dimensionless friction for different depths of textured bearings with different roughness. When the eccentricity is 0.4, the trend of dimensionless friction decreases visually with increasing weave depth from 0 to 40 μm. Influenced by the roughness, the reduction rate of the dimensionless friction is not the same when the depth of the texture is different.

### 4.3. Analysis of Frictional Wear Test

In order to investigate the effect of roughness on the lubrication characteristics of textured bearings, six sets of bearing friction pairs were machined, and their specimen parameters are shown in Table 2.

#### 4.3.1. Working Condition of Spindle Speed 200 r/min

Figure 24 shows the experimental analysis of the friction characteristics of the M2000 tester at 200 r/min for test pieces with different surface roughness. Under this condition, the friction force changes more smoothly in the range of 0.6–0.7 for roughness *Ra*. When the surface roughness reaches 0.93 μm, the friction force increases significantly with the increasing load. Overall, the coefficient of friction also increases with *Ra*.

Figure 25 depicts the effect of the depth of the surface texture on the friction characteristics. The graph clearly indicates that the increase in texture depth results in improved friction, indicating that the surface profile enhances the oil storage capacity of the friction subsurface. Theoretical calculations show that the best texture depths are 5 μm and 10 μm, and the friction characteristics of the 5 μm shank in this test were improved, but did not perform as well as those of the 20 μm and 40 μm shank. It can be seen that under this test condition, the prepared bearing friction pair should pay more attention to the mutual influence of roughness and texture in the practical application production.

#### 4.3.2. Working Condition of Spindle Speed 400 r/min

Figure 26 shows the analysis of friction characteristics for the working condition where the spindle speed is 400 r/min. When the rotational speed is increased, the change trend of the roughness of the three groups of specimens is 0.93 μm, 0.60 μm, and 0.67 μm, respectively, and under the working condition of 200 r/min, it is basically the same. However, the speed increased, resulting in an increase in the dynamic pressure effect. The friction between the fluid will also be relatively increased, and the effect of roughness on the friction also increases.

Figure 27 shows the frictional characteristics of the textured bearing at 400 r/min. When the speed increases, the dynamic pressure effect increases, and as the flow in the texture area increases, the local dynamic pressure effect is more significant in the state of flow equilibrium, so the deeper the texture depth, the lower the friction force and the better the friction characteristics.

## 5. Conclusions

(1) The theoretical calculation process to consider whether the longitudinal roughness *Ra* makes a large difference on the influence of bearing friction characteristics. If the longitudinal roughness is considered to exist in the frictional side of the dynamic pressure bearing, the maximum pressure lift of its film can be up to 47.6%;

(2) Under certain operating conditions, there is an optimal depth of texture and surface roughness Ra combination that can improve the tribological characteristics of radial dynamic pressure bearings;

(3) The laser etching process can produce a 20 μm square texture by changing the power level of the equipment to control the depth of the microfabrication. The surface texture can be prepared by a ultrasonic vibration milling process, and the surface shape with less of a roughness and uniform roughness profile can be obtained by changing the processing parameters, and the square textured surface of 5 μm can be obtained;

(4) Surface roughness has an important effect on the performance of textured bearings, and it cannot be neglected during the study of textured bearings. Under the test condition of 200 r/min in this paper, the frictional force decreased about 71% for a roughness of 0.67 μm than that for a roughness of 0.93 μm. A texture depth of 40 μm reduced the friction by 73% compared to a texture depth of 0.

## Figures and Tables

**Figure 1 micromachines-14-00577-f001:**
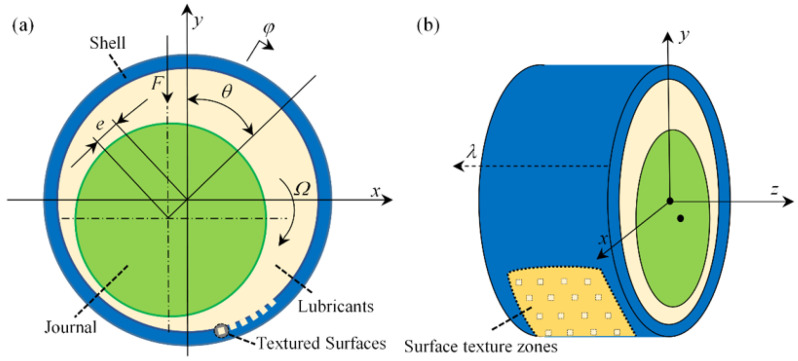
Schematic diagram of a radial dynamic pressure bearing structure (**a**) the schematic diagram of the axial direction of the textured dynamic pressure bearing, (**b**) the radial structure of the bearing.

**Figure 2 micromachines-14-00577-f002:**
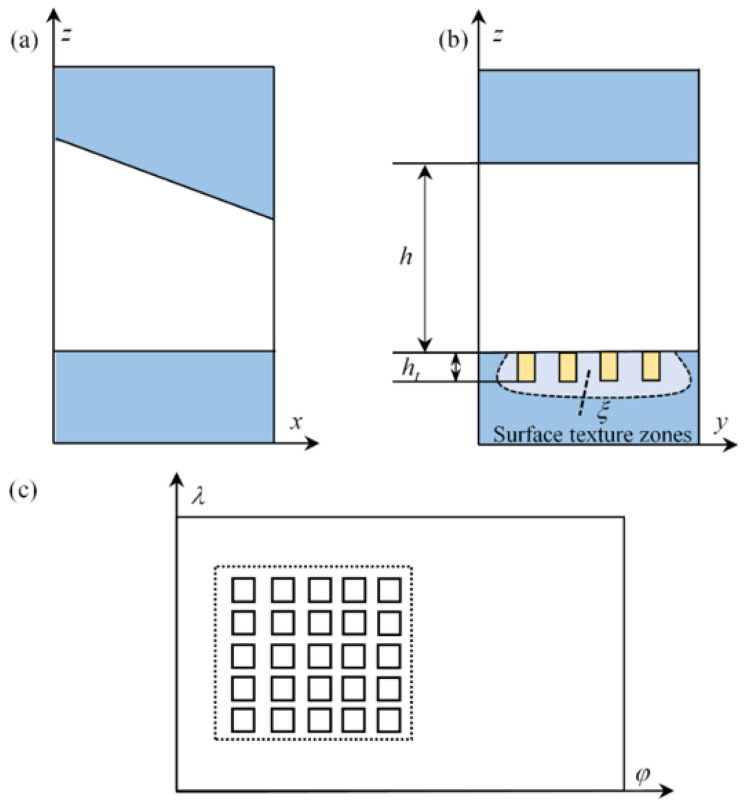
Schematic diagram of lubricant film in smooth condition (**a**) a schematic diagram of circumferential bearing gap film thickness, (**b**) a schematic diagram of axial direction film thickness, and (**c**) a schematic diagram of film expansion along the circumference.

**Figure 3 micromachines-14-00577-f003:**
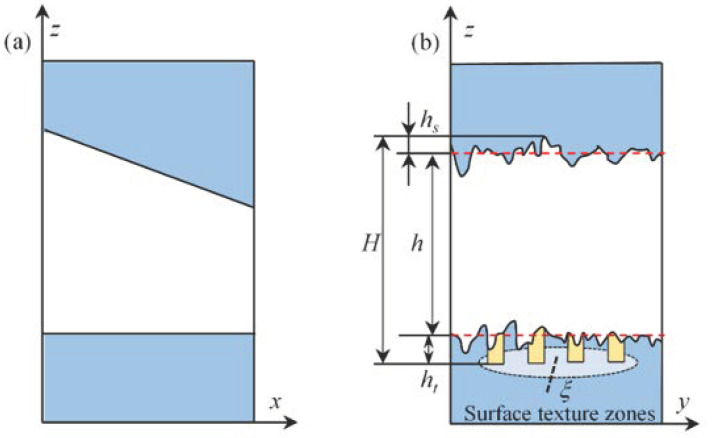
Diagram of lubricant film, taking into account roughness (**a**) a schematic diagram of circumferential bearing gap film thickness, (**b**) the schematic diagram of film thickness in the axial direction for longitudinal roughness.

**Figure 4 micromachines-14-00577-f004:**
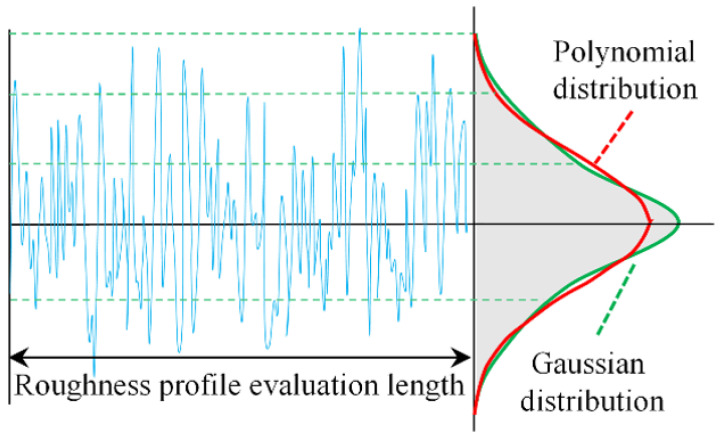
Roughness profile probability distribution.

**Figure 5 micromachines-14-00577-f005:**
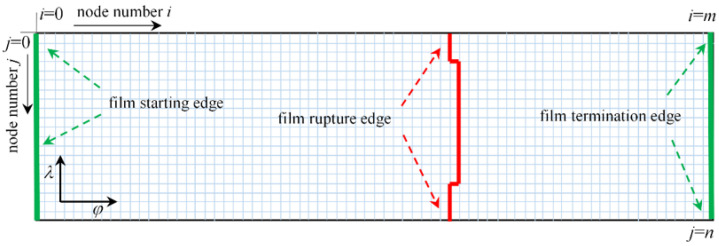
Schematic diagram of discretized film meshing.

**Figure 6 micromachines-14-00577-f006:**
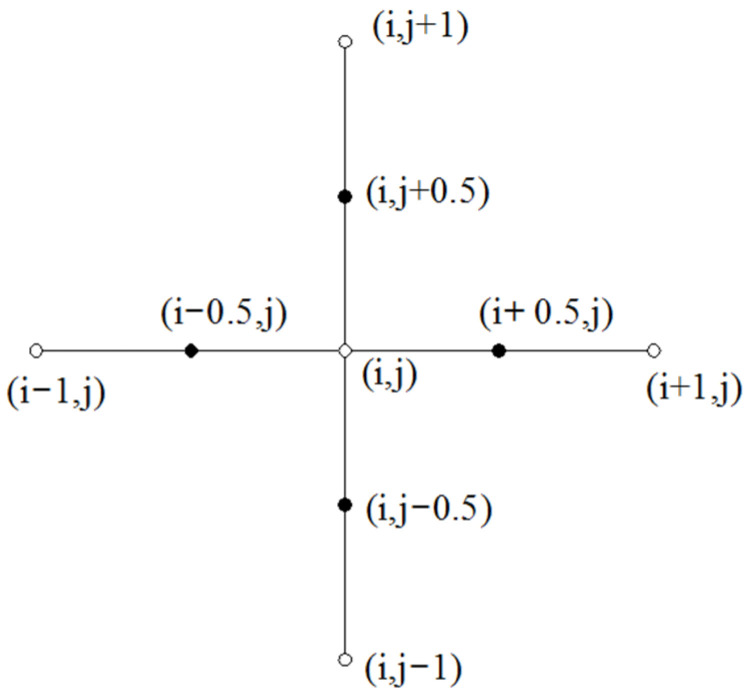
Schematic diagram of the finite difference method on a single node.

**Figure 7 micromachines-14-00577-f007:**
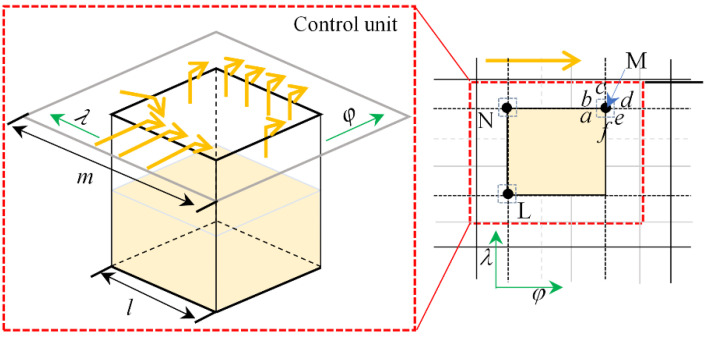
Flow balance within the texture.

**Figure 8 micromachines-14-00577-f008:**
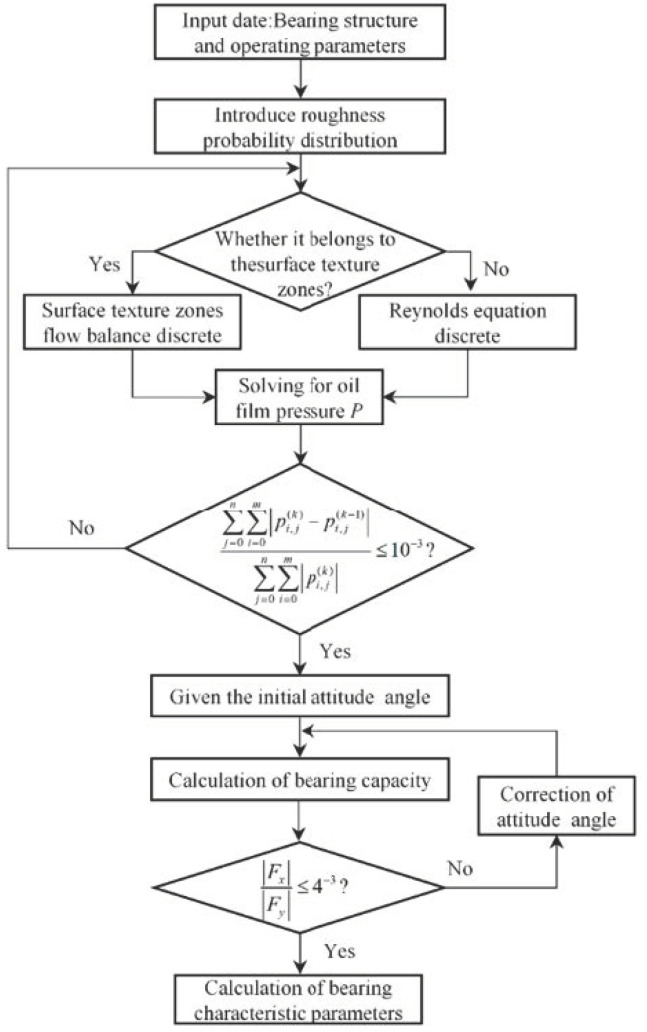
Flow chart of numerical solution.

**Figure 9 micromachines-14-00577-f009:**
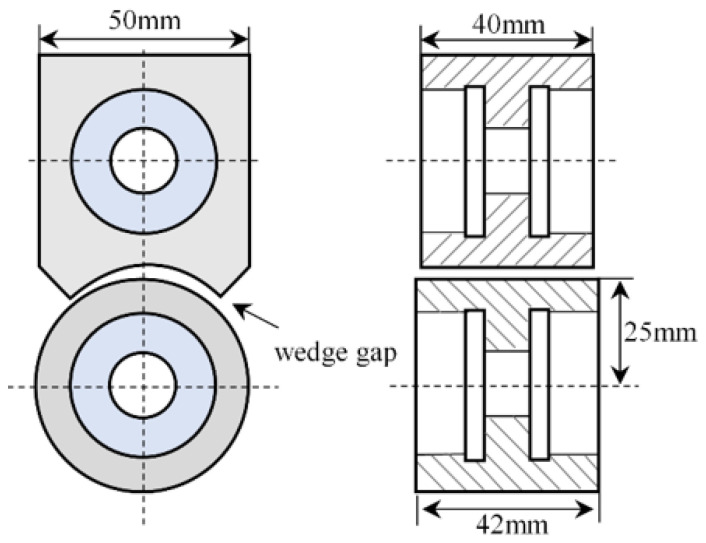
Structural parameters of the bearing friction pair.

**Figure 10 micromachines-14-00577-f010:**
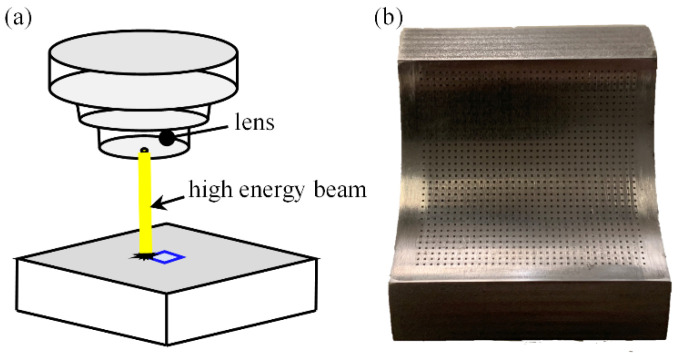
Laser etching of textured bearings (**a**) the schematic diagram of the laser etching principle, (**b**) the surface shape of the laser etched test piece.

**Figure 11 micromachines-14-00577-f011:**
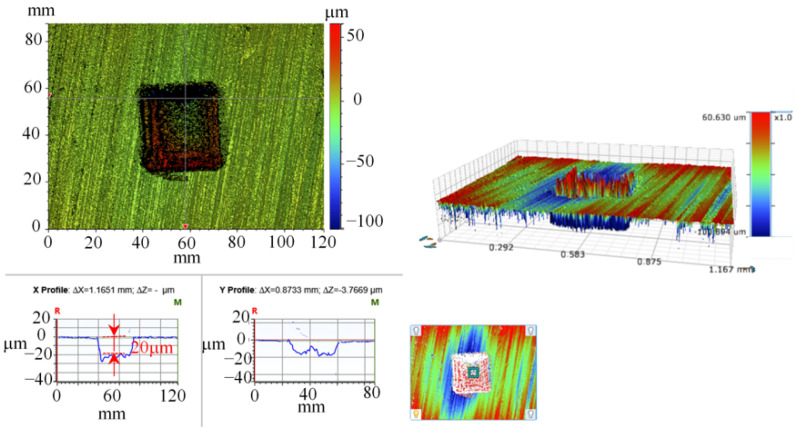
3D morphology of the texture (depth is 20 μm).

**Figure 12 micromachines-14-00577-f012:**
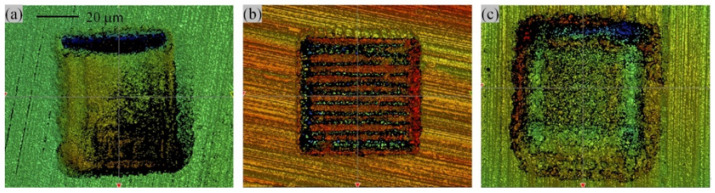
3D morphology of the texture (**a**) the surface appearance of condensation on the bottom of the texture, (**b**) the surface appearance of streaks on the bottom of the texture, (**c**) surface appearance of protrusions on the bottom of the texture.(depth is 5 μm).

**Figure 13 micromachines-14-00577-f013:**
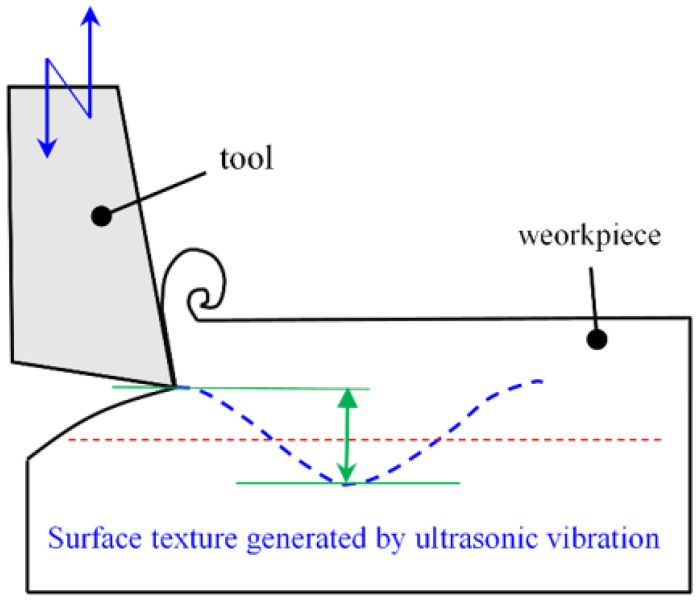
Ultrasonic vibration milling mechanism.

**Figure 14 micromachines-14-00577-f014:**
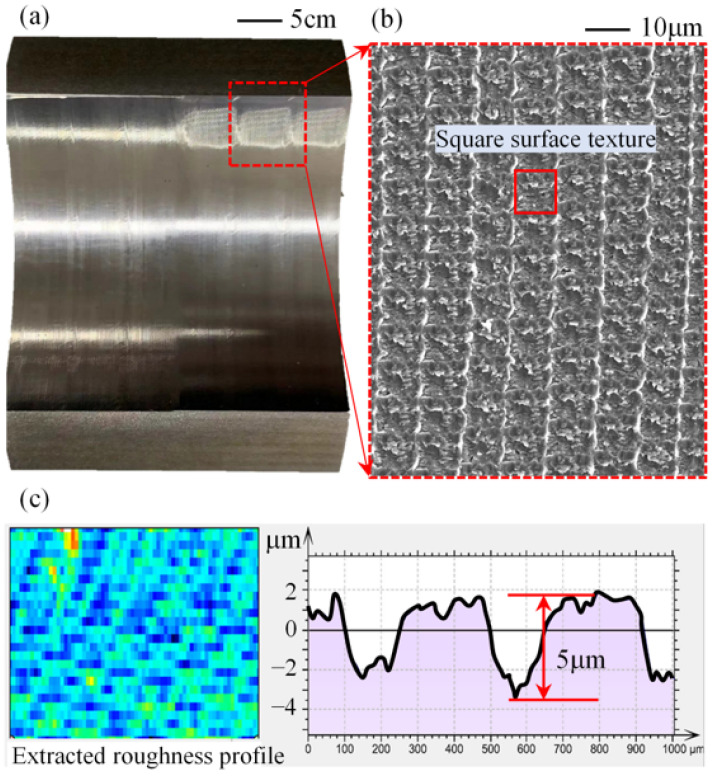
Surface texture of ultrasonic milling: (**a**) the physical view of the test piece, (**b**) the surface morphology of the test piece, and (**c**) the three-dimensional morphology of the texture.

**Figure 15 micromachines-14-00577-f015:**
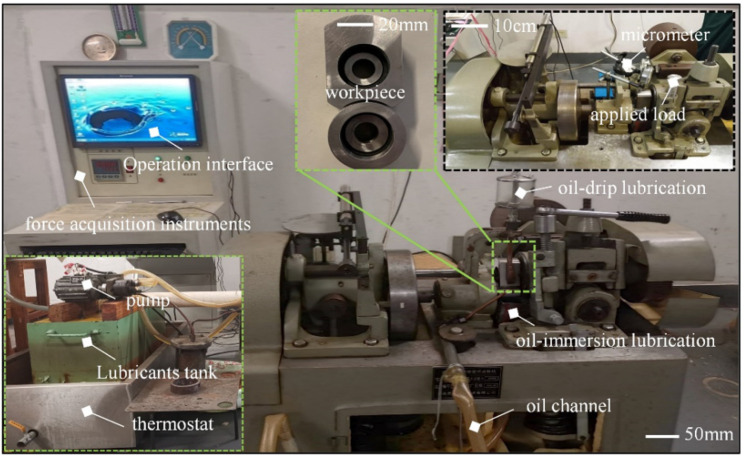
Frictional wear experimental device.

**Figure 16 micromachines-14-00577-f016:**
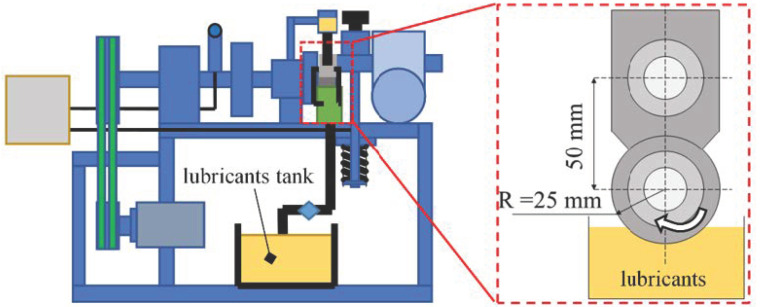
Experimental principle of frictional characteristics.

**Figure 17 micromachines-14-00577-f017:**
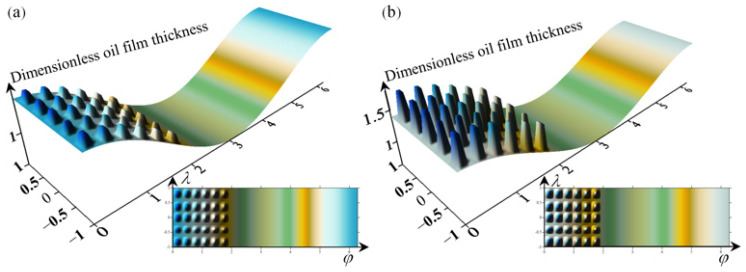
Textured bearing film unfolding diagram (**a**) a simulation of the film with a texture depth of 5 μm, (**b**) a simulation of the film with a texture depth of 20 μm.

**Figure 18 micromachines-14-00577-f018:**
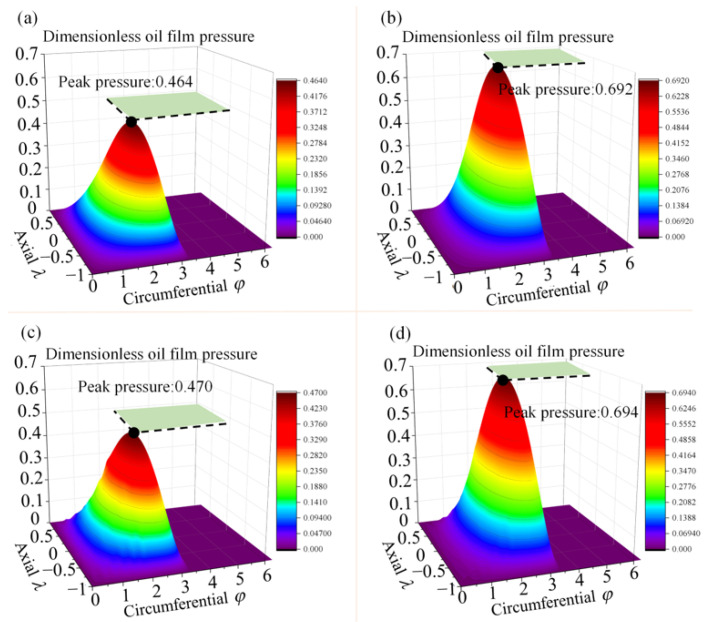
Film pressure analysis (**a**) the pressure distribution with a smooth surface, (**b**) the pressure distribution with a surface containing roughness, (**c**) the pressure distribution with a textured surface, and (**d**) the pressure distribution with coupled roughness and surface texture).

**Figure 19 micromachines-14-00577-f019:**
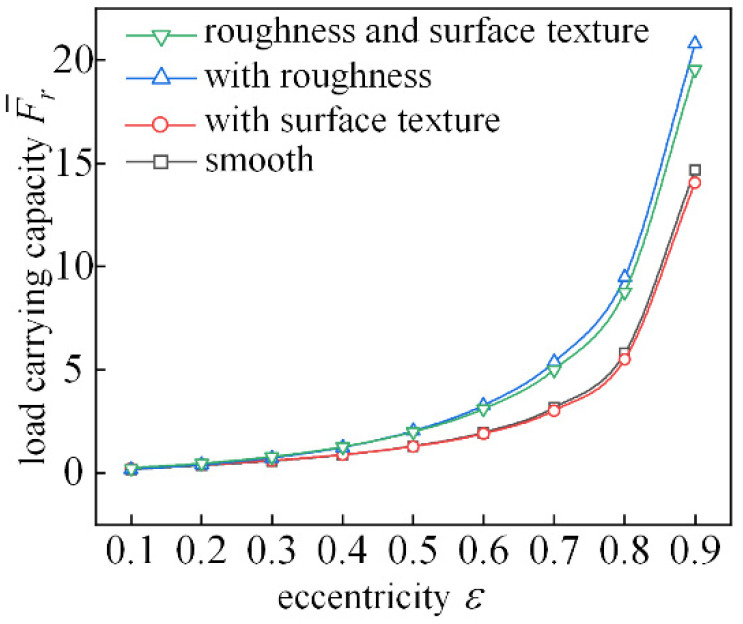
Analysis of load-bearing force characteristics changing with eccentricity.

**Figure 20 micromachines-14-00577-f020:**
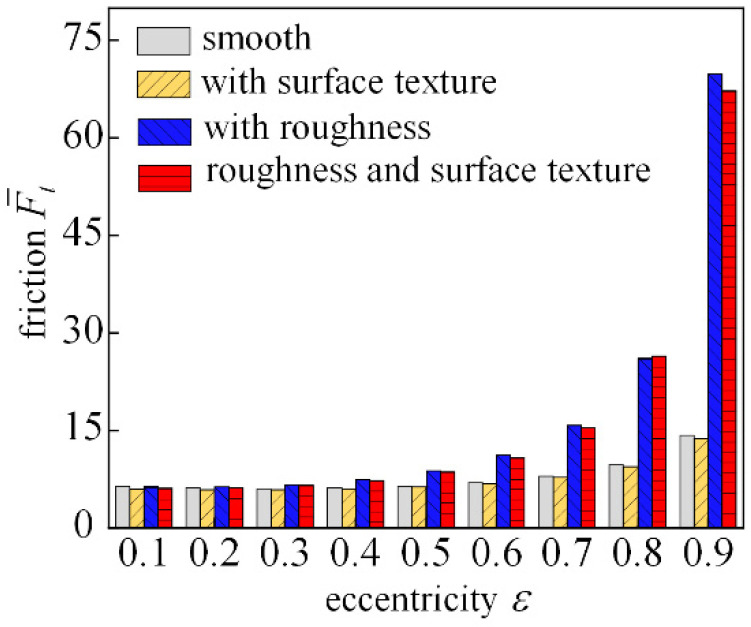
Analysis of frictional characteristics changing with eccentricity.

**Figure 21 micromachines-14-00577-f021:**
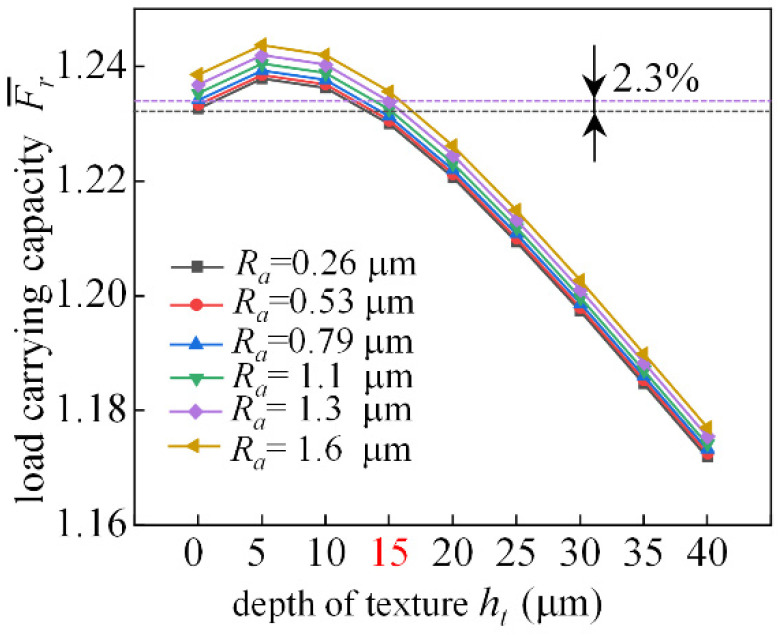
Analysis of load-bearing force characteristics changing with texture depth.

**Figure 22 micromachines-14-00577-f022:**
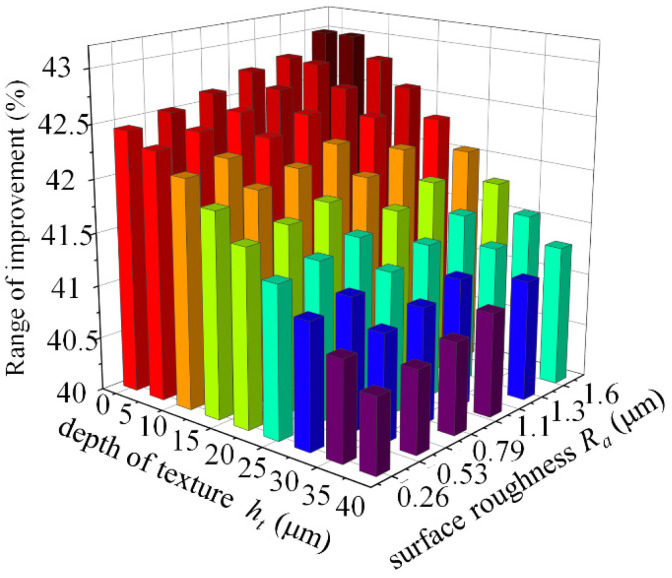
Percentage increase in load capacity.

**Figure 23 micromachines-14-00577-f023:**
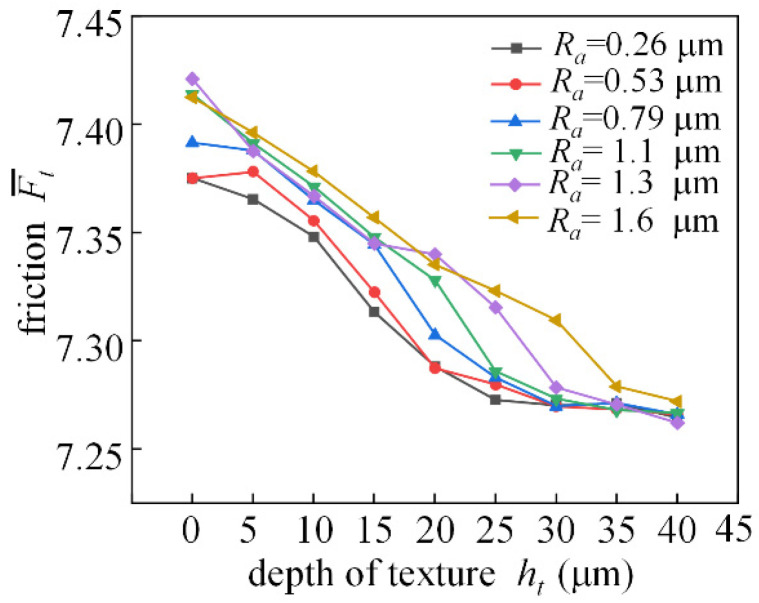
Analysis of frictional characteristics changing with texture depth.

**Figure 24 micromachines-14-00577-f024:**
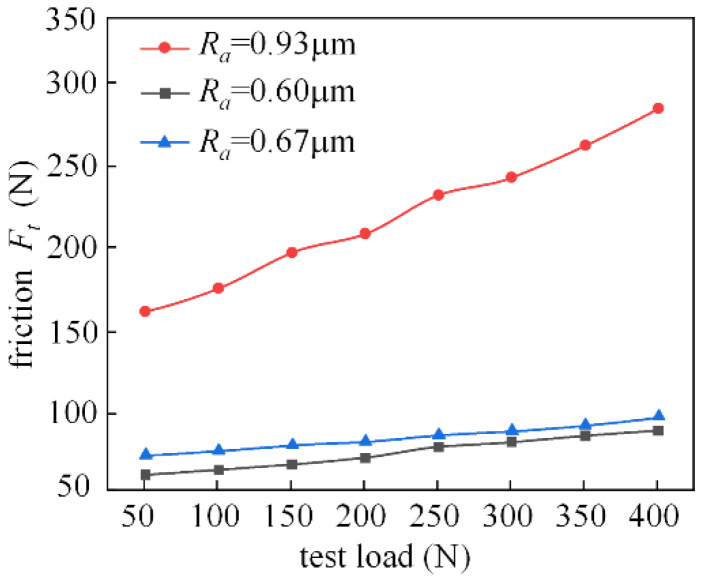
Effect of roughness on friction characteristics (the spindle speed is 200 r/min).

**Figure 25 micromachines-14-00577-f025:**
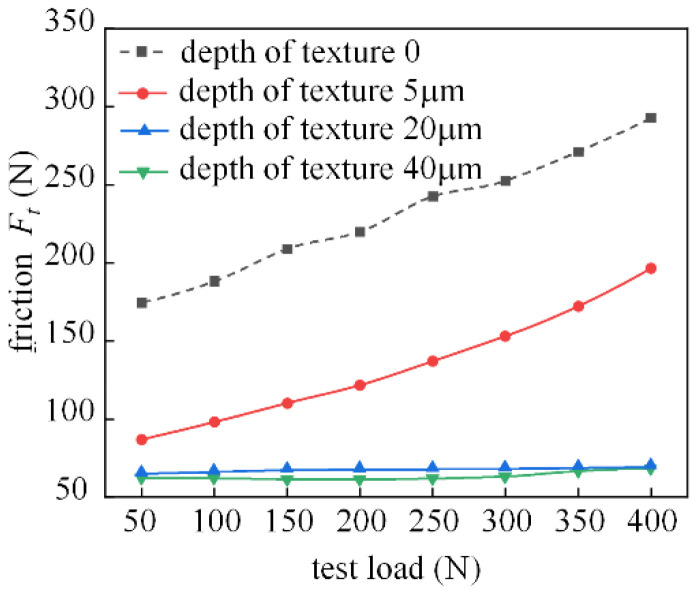
Effect of texture depth on friction characteristics (the spindle speed is 200 r/min).

**Figure 26 micromachines-14-00577-f026:**
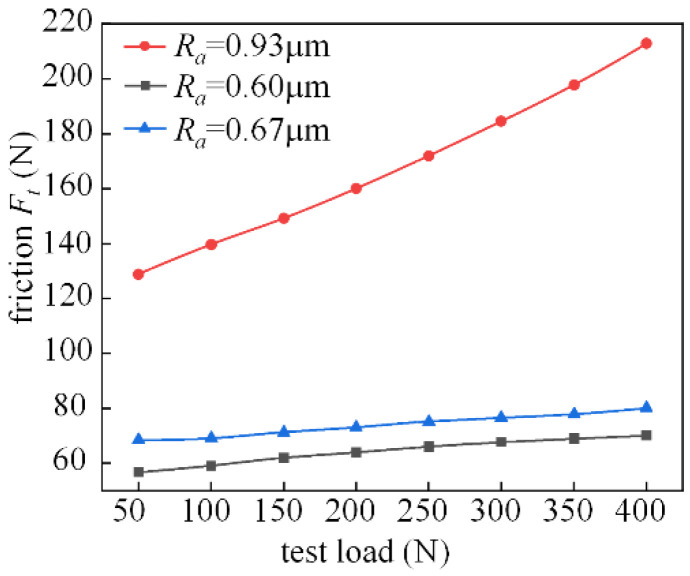
Effect of roughness on friction characteristics (the spindle speed is 400 r/min).

**Figure 27 micromachines-14-00577-f027:**
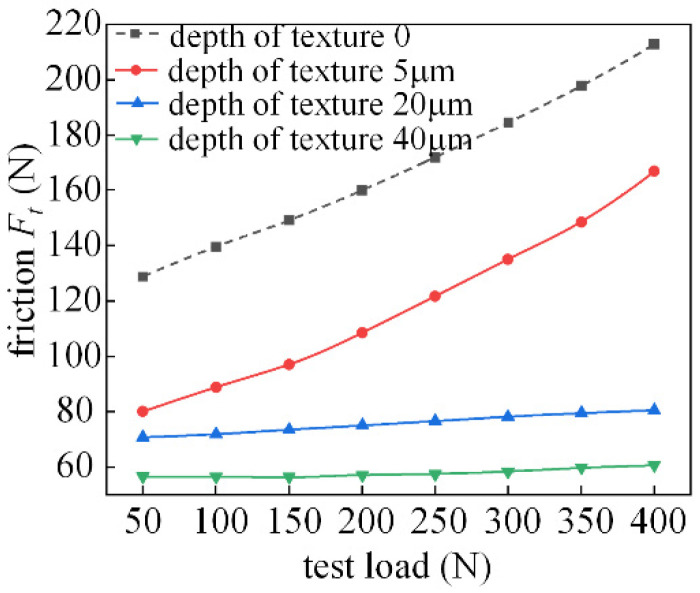
Effect of texture depth on friction characteristics (the spindle speed is 400 r/min).

**Table 1 micromachines-14-00577-t001:** Bearing working parameters.

Parameters	Symbol/Unit	Value
Length of bearing	*l*/mm	40
Radius of bearing	*r*/mm	25
Radius clearance	*c*/mm	0.05
Weight of rotor	*M*/kg	100
Eccentricity	*ε*	0.1~0.9
Lubricant density	*ρ*/kg·m^−3^	840.5
Lubricant viscosity	*μ*/Pa·s	0.04
Texture area ratio	*S*	0.36
Dimensions of the texture	*m*/μm	10
Depth of texture	*h_t_*/μm	0~40
Value of roughness	*R_a_*/μm	0~1.6

**Table 2 micromachines-14-00577-t002:** Parameters of the bearing friction pair.

Test Piece Number	Value of Roughness (μm)	Depth of Texture (μm)
No.1	0.93	-
No.2	0.6	-
No.3	0.67	-
No.4	0.51	5
No.5	0.63	20
No.6	0.47	40

## Data Availability

Due to privacy and ethical reasons, detailed data cannot be disclosed.

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
