# Peer review of "Theoretical and Experimental Study of Friction Characteristics of Textured Journal Bearing"

_micromachines, 2023, doi:10.3390/mi14030577_

Round 1
Reviewer 1 Report
The theoretical model of the square textured bearing is established and the roughness and texture are comprehensively considered and experimentally verified. Some problems should be addressed.
1) Figs 1-3, Subplots should describe in figure titles respectively.
2) What application of the bearing is aimed at? The bearing material is 42CrMo.
3) Figs19-20, considering the roughness, the bearing load capacity increases. What is the ration of textured area to rough area?
4) Figs25-27, the same scale of x- and y-axis should be used to compare.
Author Response
Point 1:Figs 1-3, Subplots should describe in figure titles respectively.
Response 1: Thank the review experts for their valuable comments, which have been revised in the article.
Point 2:What application of the bearing is aimed at? The bearing material is 42CrMo.
Response 2: Thanks for the valuable comments of the review experts, 42CrMo steel is an ultra-high strength steel with high strength and toughness, and good hardenability, which can be widely used in mechanical parts such as steam turbines and internal combustion engines.
Point 3:Figs19-20, considering the roughness, the bearing load capacity increases. What is the ration of textured area to rough area?
Response 3: Thank the review experts for their valuable comments, the bearing characteristics were calculated and analyzed with a texture percent-age of 36%.
Point 4:Figs25-27, the same scale of x- and y-axis should be used to compare.
Response 4: Thank the review experts for their valuable comments. Figure 24 has been changed. At the speed of 200r/min, the uniform scale of ordinate is 50N to 350N. The unified ordinate is 60N to 220N at 400r/min. The abscissa is from 50N to 400N If the ordinate ratio is unified, there is a large gap in the speed of 400, so the ordinate ratios of 200 r/min and 400 r/min are unified, so only the coordinate ratios of 200 and 400 are unified.

Reviewer 2 Report
The manuscript titled as“Theoretical and experimental study of friction characteristics of textured journal bearing” is mainly researched the effects of surface roughness and texture depth on friction and oil film pressure and load . The Reynolds equation containing longitudinal roughness is established for journal bearing and obtained the bearing load and friction characteristics. In order to obtain the texture surface , the processing method of laser etching and ultrasonic vibration milling was used. Then the processed sample is put on the M2000 tribological wear testing machine for experimental research. The COXEM EM-30plus benchtop SEM and a Bruker Nano Fnc white light interference instrument were used to photograph the surface roughness and texture depth he bearing surface morphology. The surface roughness can substantially increase the oil film pressure. The depth of the texture affects friction and bearing capacity. This research has high research value. But there are some contents to be modified . I am recommending major revision of this article because of the following issues:
1. English language should be on much higher level.Please check2th page row 69-70,93; 3th page row 108;4th page row123-124,133-134;5th page row147,148; 7th page row185-186; 10th page row234-237,249-252; 12th page row280; 13th page row296-297; 14th page row324-325; 16th page row369,370; 17th page row393,374;
2. 4th page row 109-115 ,8th page row212-216, 9th page row223-227;. Sentence is very very long, ambiguous. It should be shortened and clarified.
3. Please check the number of formula in 6th page. There are two formulas numbered 9.
4. Please check3th page row100,108,The c symbol is not explained on row 100.
5. Please check the Figure9 and 21, There are dimensions not marked in Figure 9 .2.3% in Figure 21 is not explained .
6. Please check the all the Figure in the paper and please keep the format of the first letter consistent for the description of horizontal and vertical coordinates.
7. Details about the bearing film measurements should be given, especially the measurements error about the test equipment.
8. The friction characteristics are not only related to friction, but also to other parameters such as friction coefficient .You'd better give an explanation.
9. The description in Figure 25 only describes the relationship between texture depth and oil, which is not convincing
10. In order to explain the effect of speed on roughness and friction on page 19, only two speeds parameters are not enough.
11. There is no comparison between the theoretical calculation model and the test data, so there is no way to explain the correctness of the theoretical model.
12. The conclusions 3 and 4 in the article need to be rewritten, the paper only introduces the sample processing method, but does not introduce the processing technology in detail, so it is not recommended to write this into the conclusion. The conclusion 4 is not closely related to the test results.

Author Response
Point 1: English language should be on much higher level. Please check2th page row 69-70,93; 3th page row 108;4th page row123-124,133-134;5th page row147,148; 7th page row185-186; 10th page row234-237,249-252; 12th page row280; 13th page row296-297; 14th page row324-325; 16th page row369,370; 17th page row393,374;
Response 1: Thank the experts for their valuable comments. We have checked the English language and revised it in the text.
Point 2:4th page row 109-115 ,8th page row212-216, 9th page row223-227;. Sentence is very very long, ambiguous. It should be shortened and clarified.
Response 2: Thank the experts for their valuable comments. These sentences have been revised in the text.
Point 3:Please check the number of formula in 6th page. There are two formulas numbered 9.
Response 3: Thank the experts for their valuable comments. The formula number has been modified in the text.
Point 4:Please check3th page row100,108,The c symbol is not explained on row 100.
Response 4: Thank the experts for their valuable comments. The c symbol has been explained in the text.
Point 5:Please check the Figure9 and 21, There are dimensions not marked in Figure 9 .2.3% in Figure 21 is not explained .
Response 5: Thank the review experts for their valuable comments. These questions have been revised in the text。
Point 6:Please check the all the Figure in the paper and please keep the format of the first letter consistent for the description of horizontal and vertical coordinates.
Response 6: Thank the review experts for their valuable comments, which have been checked and revised in the paper.
Point 7: Details about the bearing film measurements should be given, especially the measurements error about the test equipment.
Response 7: Thank the review experts for their valuable comments, as shown in the figure above, the angle of 14.9838 ° is the initial offset angle of the test, 0.0202 and 0.05 are the clearance between the oil inlet and outlet, and the oil film of the bearing cannot be measured directly, but can be given indirectly by using the offset angle. Later, the oil film will also change when the initial offset angle is changed by loading.
Point 8:The friction characteristics are not only related to friction, but also to other parameters such as friction coefficient . You'd better give an explanation.
Response 8: Thank the review experts for their valuable comments. In this test, M2000 friction and wear tester was used to verify the friction characteristics. The friction characteristic data collected by the sensor is shown in the figure above. Because of the direct conversion relationship between friction and friction coefficient, in order to facilitate the presentation of the results, the friction is taken as the result to explain, and the obtained results are consistent with the change ratio of friction coefficient. Therefore, the result is finally presented with friction as the change form of friction characteristics.
Point 9:The description in Figure 25 only describes the relationship between texture depth and oil, which is not convincing
Response 9: Thank the review experts for their valuable opinions. The smooth bearing surface can reduce the bearing friction by adding texture. Its working principle is that on the one hand, the texture can store a certain amount of abrasive debris, on the other hand, the texture can store a certain amount of lubricating oil, and within a certain range of texture depth, the deeper the texture, the more lubricating oil stored inside, making the oil supply more sufficient during the bearing operation, More adequate lubrication. The test results also verify this point, and the friction decreases gradually with the increase of depth.
Point 10:In order to explain the effect of speed on roughness and friction on page 19, only two speeds parameters are not enough.
Response 10: Thank the review experts for their valuable opinions. It is difficult to process the complete textured dynamic sliding bearing and the data measurement error is large. In order to simplify the test, we use a quarter of the bearing bush and test it on the M2000 friction and wear test bench (for the convenience of processing and the accuracy of data measurement). The main shaft of the test platform has only two parameters, 200r/min and 400r/min, which are set at the factory. The change of bearing friction with the increase of load at two different rotational speeds is tested and compared with the theoretical calculation results. The test results are consistent with the calculated results, which can prove the correctness of the theoretical model. In the theoretical calculation, we can use different rotational speeds for calculation. However, it is undeniable that the test speed parameters are too few. In the subsequent research process, we will optimize the test equipment and add more speed parameters.
Point 11:There is no comparison between the theoretical calculation model and the test data, so there is no way to explain the correctness of the theoretical model.
Response 11: Thank the review experts for questioning the conclusion. The author himself also noticed the shortcomings of the test during the process of test verification. In the process of preliminary planning of the test, it is really necessary to consider the combination of test results verification and theoretical results. Because there are many defects in the design process of the test, the ideal results cannot be achieved. For example, the bearing pads used in this test are part of the pads, as shown in the figure below. The wedge gap formed between the shaft and the bearing pad will also affect the dynamic pressure effect of the oil film, resulting in systematic errors in the friction characteristics. The main shaft speed of the testing machine is 200r/min and 400r/min respectively. These two speeds are quite different from the actual working conditions, which will also cause system error. In order to ensure the formation of dynamic pressure oil film, the test load is not easy to be too large. If the applied load is too large, the friction state will change, mixed friction or dry friction will occur, and the workpiece surface will be scratched, so the prepared surface texture will fail, as shown in the following figure. In view of the limitations of the above test equipment, the test results will have a large error with the theoretical calculation. Therefore, during the analysis and discussion of the test data, only the change trend of the friction characteristics of textured bearings was verified, and the effects of different roughness and texture depth on the friction characteristics were compared.
In the problem of the correctness of the theoretical model, this paper establishes a coupling model considering the two factors of surface texture and random roughness, discretizes the equation through the finite difference method, and compiles the C language program. In previous studies, we have established a textured bearing model and a bearing model considering the change of roughness, and compared the theoretical calculation results with other literature results, the error is within the allowable range, which proves the correctness of our theoretical results.
Point 12: Conclusions 3 and 4 in the text need to be rewritten, and the paper only introduces the sample processing method, but does not introduce the processing technology in detail, so it is not recommended to write the conclusion. Conclusion 4 is not closely related to the test results.
Response 12: Thank you to the evaluation experts for their valuable comments. This is because it is difficult to achieve the same roughness values during workpiece machining and it is difficult to measure the non-textured areas of textured bearings. The average roughness value of a textured surface becomes larger, making it difficult to compare the test results of textured bearings with roughness in testing. Conclusions 3 and 4 have been revised in the text.

Round 2
Reviewer 2 Report
The manuscript titled as“Theoretical and experimental study of friction characteristics of textured journal bearing” has high research value. But there are some contents to be modified .
1. English language should be modified .Please check2th page row 95-96;
2. Please check the formula of 16 and 18 in 9th page. The meaning of parameters in the formula are not explained.

Author Response
Point 1:English language should be modified .Please check2th page row 95-96;
Response 1: Thank the experts for their valuable comments. We have checked the English language and revised it.
Point 2:Please check the formula of 16 and 18 in 9th page. The meaning of parameters in the formula are not explained.
Response 2: Thank the experts for their valuable comments. The parameters in the formula have been explained in the paper.
